# Isoflavone Protects the Renal Tissue of Diabetic Ovariectomized Rats via PPARγ

**DOI:** 10.3390/nu14132567

**Published:** 2022-06-21

**Authors:** Adriana Aparecida Ferraz Carbonel, Rafael André da Silva, Luiz Philipe de Souza Ferreira, Renata Ramos Vieira, Ricardo dos Santos Simões, Gisela Rodrigues da Silva Sasso, Manuel de Jesus Simões, José Maria Soares Junior, Patrícia Daniele Azevedo Lima, Fernanda Teixeira Borges

**Affiliations:** 1Structural and Functional Biology Graduate Program, Paulista School of Medicine, Federal University of São Paulo (EPM/UNIFESP), 740 Edifício Lemos Torres—2° andar, Vila Clementino, São Paulo 04023-900, SP, Brazil; luiz.philipe@unifesp.br (L.P.d.S.F.); renata.vieira24@unifesp.br (R.R.V.); gisela.morf@hotmail.com (G.R.d.S.S.); mjsimoes@unifesp.br (M.d.J.S.); 2Department of Gynecology, Paulista School of Medicine, Federal University of São Paulo (EPM/UNIFESP), São Paulo 04023-900, SP, Brazil; 3Biosciences Graduate Program, Institute of Biosciences, Letters and Exact Sciences, São Paulo State University (IBILCE/UNESP), São José do Rio Preto 15054-000, SP, Brazil; rafaels@usp.br; 4Department of Obstetrics and Gynecology, Medicine Faculty of University of São Paulo (FMUSP), São Paulo 05403-911, SP, Brazil; ricardo.simoes@amb.org.br (R.d.S.S.); jsoares415@hotmail.com (J.M.S.J.); 5Queen’s Cardiopulmonary Unit (QCPU), Queen’s University, Kingston, ON K7L 3J9, Canada; pdal@queensu.ca; 6Department of Medicine, Nephrology Division, Paulista School of Medicine, Federal University of São Paulo (EPM/UNIFESP), São Paulo 04023-900, SP, Brazil; ft.borges@unifesp.br; 7Interdisciplinary Graduate Program in Health Sciences, Cruzeiro do Sul University, São Paulo 01506-000, SP, Brazil

**Keywords:** isoflavone, kidney, PPARγ, diabetes, postmenopausal

## Abstract

Diabetes associated with post-menopause is related to a worse condition of kidney disease. Taking into consideration that this disorder may be regulated by estrogenic mediators, we evaluated the renal protective effect of isoflavone. We investigated the role of the PPARγ in the pathogenesis of the disease. For this study, we used diabetic female rats in a postmenopausal model through ovariectomy. The animals were treated with isoflavone or 17β-estradiol. A dosage was administered to bring on blood glycemia, and through immunohistochemistry, we evaluated the immunoreactivity of PPARγ in the endometrium and renal tissue. We analyzed the immunoreactivity of renal injury molecule KIM-1 and the collagen and glycogen densities in the kidney. Through bioinformatics analysis, we observed *PPARγ* and *COL1A1* gene expression under the influence of different glucose doses. In particular, we observed that isoflavone and 17β-estradiol regulate blood glycemia. Renal injury was inhibited by isoflavone, observed by a reduction in KIM-1, along with glycogen accumulation. These benefits of isoflavone may be associated with PPARγ overexpression in the kidneys and endometrium of diabetic ovariectomized rats.

## 1. Introduction

Type 1 diabetes (T1D) has high rates of morbidity and mortality and an increasing incidence. From the moment of diagnosis, the management of diabetes mellitus (DM) places an enormous burden on the patient, with severe limitations, changes in lifestyle, and organ dysfunction [1]. Diabetic nephropathy (DN) is one of the leading causes of kidney disease worldwide, and being in the post-menopause stages of life contributes to a worse prognosis [2,3,4], as postmenopausal sex hormone levels affect the renal/vascular physiology [5].

Estrogen prevents the development and progression of kidney disease and improves postmenopausal symptoms, especially in diabetes, the main risk factor for chronic kidney disease. In addition, women with lower endogenous exposure to estrogen have a higher risk of kidney disease [6]. Isoflavone is known for its estrogenic role [7]. The beneficial effect of isoflavone has been demonstrated in an animal model of diabetes [8] and in humans [9]. The ingestion of soy protein, total isoflavones, daidzein, and genistein is associated with lower risks of DM2 in a dose-dependent manner in women, though this effect is not evident in men [9]. These observations may be population-dependent, as in a cross-section of North American subjects, isoflavone was shown to be effective for men and women with DM2 [10]. Nevertheless, few studies have demonstrated the effect of isoflavone on T1D, especially in renal injury [11,12].

Peroxisome proliferator-activated receptor gamma (PPARγ) has been closely associated with diabetes, menopause, and kidney disease. Due to its role in glucose regulation, PPARγ agonists are suggested to ameliorate the problems of diabetes [13]. Our group demonstrated that isoflavone positively regulates PPARγ. In addition, work from the lab demonstrated beneficial effects on diabetic conditions and in obese animals [12,13,14]. We previously demonstrated that isoflavone protects from kidney injury in obese animals via PPAR-γ [14], prevents bone loss in ovariectomized diabetic animals [15], and can attenuate the effects of vulvovaginal atrophy in women [16]. Taking into account the protective function of PPAR-γ in kidney injury [17,18,19,20], the objective of this work was to evaluate the effect of ovariectomy and the consequent gonadal failure on the development of DM1-induced kidney injury, as well as the contribution of soy isoflavone via PPAR-γ in this experimental animal model. Our hypothesis is that gonadal failure potentiates renal injury induced by chronic hyperglycemia and that isoflavone treatment attenuates this condition by PPAR-γ induction. 

Our data revealed that isoflavone protects the kidney against gonadal failure associated with DM1 damage. Increased protein and gene expression of PPAR-γ was evidenced under the influence of isoflavone. In addition, we demonstrate for the first time that PPAR-γ reduction occurs dose-dependent on glucose. On the other hand, we have not clearly demonstrated the consequences of ovariectomy in the renal injury development, and other findings are necessary to confirm this hypothesis.

## 2. Materials and Methods

### 2.1. Animals, Experimental Design, and Treatments

Our study sample consisted of 60 adult, virgin female rats (Rattus norvegicus Albinus), three months of age, weighing between 180 and 210 g. The in vivo experimental protocol was approved by the Ethics Committee (Research Ethics Committee 001676/09) of the Federal University of São Paulo. The animals were donated by the Center for the Development of Experimental Models of the Federal University of São Paulo (CEDEME). All animals were kept in plastic cages with controlled light (12/12 h light/dark cycles) and temperature (22–24 °C), received water and food ad libitum, and were fed a specific proportion of soy-free food (Formula Labina Especial for rodents), developed by Agribrands-Purina, São Paulo, Brazil.

After a one-week adaptation period, the rats were anesthetized with an intraperitoneal injection of 10% ketamine (0.08 mL/100 g/p.p; RompunVR, São Paulo, Brazil) and 2% xylazine (0.04 mL; KetalarVR, São Paulo, Brazil). They then underwent bilateral ovariectomy immediately after surgery. One day after bilateral ovariectomy, type 1 diabetes in animals was induced by a single-dose intraperitoneal injection of freshly prepared 60 mg/kg streptozotocin (STZ—Sigma-Aldrich, St. Louis, MO, USA) dissolved in a 0.1 mol/L citrate buffer (pH = 4.8). Hyperglycemia was confirmed 48 h after the STZ injection by measuring the tail vein blood glucose levels using a blood glucose monitoring kit (Accu Check ACTIVE Roche^®^). Only animals with mean plasma glucose levels > 250 mg/dL were accepted as having diabetes mellitus [21].

Treatment with soy isoflavone and 17β-estradiol was performed for 30 days after T1D induction. Was used the concentrated extract of soy isoflavones NovasoyVR (Archer Daniels Midland, Decatur, IL, USA) containing 40% of total isoflavones in the ratio of 1.3:1:0.3 genistein: daidzein: glycitein, respectively, 7–12% protein, 4% ash, and 6% moisture, and the remaining 41% consists of the other soy phytocomponents. In addition, we used 17β-estradiol (Sigma-Aldrich). Administration of soy isoflavones or 17β-estradiol was initiated 30 d after the ovariectomy. Soy isoflavones (150 mg/kg) were administered by gavage, and 17β-estradiol (10 mg/kg) was administered subcutaneously. 

After 30 days of experimentation, all animals were anesthetized with xylazine (RompunVR, SP, Brazil) 15 mg/kg associated with ketamine (KetalarVR, SP, Brazil) 30 mg/kg, intraperitoneally. A portion of the endometrium and kidney was immediately immersed in liquid nitrogen e and then stored in the −80 °C freezer for qPCR assay. Another portion of the endometrium and kidney were immersed in paraformaldehyde 4% for histological processing.

### 2.2. Periodic Acid–Schiff Staining and Immunohistochemistry

Paraffin sections were subjected to alcohol and xylene gradient solutions, antigen retrieval, protein block, and incubation with primary antibodies against PPAR-γ polyclonal antibody (1:200, Abcam, MA, USA) and kidney injury molecule 1 (KIM-1) (1:500, rabbit IgG, Sino Biological, Beijing, China) overnight at 4 °C. After this time, the sections were incubated with streptavidin-peroxidase for 30 min (Dako, CA, USA). Periodic acid–Schiff staining was carried out according to the manufacturer’s instructions. The obtained microscope images were calculated using Leica DFC 310 FX image analysis software (Leica do Brasil Importação e Comércio Ltd., São Paulo, Brazil) and are expressed as the percentage/stained area.

### 2.3. Collagen Fiber Density

To evaluate the birefringence pattern of collagen fibers in the kidney, two sections per animal (100 mm distance between sections) were selected and studied using the picrosirius red method [21,22]. PS-stained kidney sections were also analyzed under polarized light to differentiate collagen fibers type I (yellow and red). For that, randomly selecting four non-overlapping areas in each slide were considered at 40× magnification using an Axiolab 2.0 optical microscope (Zeiss, Germany). The quantification of the area occupied by the collagen fibers was performed using the Image J software. Densitometry analysis data were obtained as arbitrary units between 0 and 255. A mean of values from the two sections/animals was obtained to calculate the mean of each group [23].

### 2.4. Gene Expression Analysis by Reverse-Transcription Quantitative PCR (RT-qPCR)

According to the manufacturer’s instructions, total RNA from the endometrium and kidney was isolated using TRIzol Reagent (Thermo Fisher Scientific Inc., Waltham, MA, USA). Total RNA was treated with DNase (RQ1 RNase-Free DNase, Promega, Madison, WI, USA) to prevent genomic deoxyribonucleic acid contamination, and the RNA pellet was resuspended in RNase-free water. For mRNA expression analysis, the total RNA (1 μg) of the endometrium and kidney samples was reverse transcribed using SuperScript III RT (Thermo Fisher Scientific Inc., Waltham, MA, USA) according to the manufacturer’s protocol. The PCR product was amplified from cDNA using QuantiFast SYBR Green PCR kit (Qiagen) and specific primers for rats (Table 1); β-actin was used as endogenous control. Amplification and detection were performed using the real-time PCR system ABI 7500 (AppliedBiosystem, Waltham, MA, USA). mRNA data are reported as normalized expression, calculated based on the 2^−ΔΔC^_T_ method. All reactions were performed in triplicate.

### 2.5. Bioinformatic Analysis

A study containing publicly available transcriptome data was selected from the Gene Expression Omnibus (GEOR) repository: (GSE168072 https://www.ncbi.nlm.nih.gov/geo/query/acc.cgi?acc=GSE168072, accessed on 3 May 2022). Primary human renal tubular epithelial (hRTE) cells. The cells were grown using serum-free conditions. The growth formulation consisted of a 1:1 mixture of Dulbecco’s modified Eagles’ medium and Ham’s F-12 growth medium supplemented with selenium (5 ng/mL), insulin (5 μg/mL), transferrin (5 μg/mL), hydrocortisone (36 ng/mL), triiodothyronine (4 pg/mL), and epidermal growth factor (10 ng/mL). The cells were fed a fresh growth medium every 3 days, and at confluence, the cells were sub-cultured using trypsin–ethylenediaminetetraacetic acid (0.05%, 0.02%). For use in experimental protocols, cells were subcultured at a 1:2 ratio, allowed to reach confluence (7 days following subculture), and fed with 5.5, 7.5, 11, or 16 mM glucose for 7 days, followed by sub-culturing the cells for 3 passages (P3) in media containing 5.5, 7.5, 11, or 16 mM glucose. Typically, the cells were passaged, 1:2 for three passages before the initiation of the experiment [24]. The datasets were analyzed individually using license-free algorithms from the GEO2R tool (available at http://www.ncbi.nlm.nih.gov/geo/geo2r/, accessed 3 May 2022), which allows users to compare different groups of samples in a GEO series to examine differentially expressed genes according to experimental conditions. GEO2R was applied to detect *PPAR-γ* and Collagen type I alpha 1 chain (*COL1A1*) genes among different experimental conditions. The *p*-values of gene expression after Log2 transformation were used to calculate the Z-score (individual value—population average/population standard deviation).

### 2.6. Statistical Analysis

Statistical analyses were performed with the GraphPad Prism (v9.1) software. The Kolmogorov–Smirnov or Shapiro–Wilk test was used to determine the normality of the data. The data of the experimental groups were compared using a one-way analysis of variance (ANOVA), followed by the application of the Kruskal–Wallis test for nonparametric samples. *p*-values less than 0.05 were considered statistically significant.

## 3. Results

### 3.1. Glucose Modulation by Isoflavone

Firstly, we confirmed our diabetes model (Figure 1). We observed the glycemic level, where DM1 and DM1 + OVX animals compared with the control group (**** *p* < 0.0001) and only OVX (** *p* < 0.01), respectively, showed hyperglycemia. Relevant to this work, we observed isoflavone-mediated glucose modulation; we demonstrated that isoflavone but not 17β-estradiol can reduce glycemia in DM1 and DM1 + OVX groups (* *p* < 0.05).

### 3.2. Expression of PPAR-γ in Isoflavone-Induced Endometrium 

Through immunohistochemistry analysis, we assessed the immunoreactivity of PPAR-γ (Figure 2a–f). Through densimetry of the PPAR-γ immunoreactivity (Figure 2g), we observed that isoflavone is a potent inducer of PPAR-γ even under the influence of DM1 + OVX when compared with the control (* *p* < 0.05), OVX (*** *p* < 0.001), and untreated DM1 + OVX (**** *p* < 0.0001) groups. In addition, a higher immunoreactivity of PPAR-γ was observed in the DM1 + OVX group treated with 17β-estradiol compared to the untreated DM1 + OVX group (*** *p* < 0.001). However, in the evaluation of PPAR-γ gene expression, we observed that only the DM1 + OVX group treated with isoflavone could positively modulate PPAR-γ mRNA compared to the untreated DM1 + OVX group (* *p* < 0.05) (Figure 2h).

### 3.3. Expression of PPAR-γ Renal Tissue

Similar to the endometrium, we observed modulation of PPAR-γ in our experimental groups (Figure 3). By immunohistochemical analysis, we observed that the DM1 + OVX group treated with isoflavone presented a higher immunoreactivity of PPAR-γ when compared with the OVX group (* *p* < 0.05), untreated DM1 + OVX group (* *p* < 0.05), and DM1 + OVX group treated with 17β-estradiol (* *p* < 0.05) (Figure 3g). Gene expression evaluation revealed that the DM1 + OVX group treated with isoflavone or 17β-estradiol overexpressed PPAR-γ when compared with the untreated DM1 + OVX group (**** *p* < 0.0001) (Figure 3h). In addition, we demonstrated, in transcriptome analysis of human renal distal tubule cells, that there is a dose-dependence of the glucose concentration on the PPAR-γ expression. From 7.5 mM glucose (*p* *** < 0.001), there is a reduction in PPAR-γ expression; this decline occurs in a concentration-dependent manner at 11 mM (*p* *** < 0.001) and 16 mM (**** *p* < 0.0001) glucose concentrations compared to the control (Figure 3i).

### 3.4. Assessment of Renal Injury

To analyze the possible protective role of isoflavone through the positive regulation of PPAR-γ, we evaluated the marker KIM-1 to assess renal injury (Figure 4). Firstly, we observed that the DM1 group exhibited greater renal injury, observed by increased KIM-1 immunoreactivity compared to the control group (**** *p* < 0.0001) and even when compared to the OVX group (** *p* < 0.01) (Figure 4g). We demonstrated that treatment with isoflavone, but not 17β-estradiol, in the DM1 + OVX group could reduce the renal injury marker KIM-1 when compared with the untreated DM1 + OVX group (** *p* < 0.01) (Figure 4g).

### 3.5. Evaluation of Type-I Collagen in Renal Tissue

To evaluate the ability of isoflavone to modulate the extracellular matrix of the actual tissue, in particular, type I collagen, through the birefringence technique to observe the collagen in the extracellular matrix, we observed that DM1 + OVX induction increases the amount of collagen in the renal tissue when compared with the control group (**** *p* < 0.0001) and OVX alone (*** *p* < 0.001) (Figure 5). Treatment with isoflavone and 17β-estradiol in the DM1 + OVX group maintained a collagen profile similar to the control group (Figure 5). To corroborate these observations, we performed a transcriptome analysis, where we demonstrated that a high glucose concentration (16 mM) promotes COL1A1 expression when compared to the control (5 mM) (* *p* < 0.05) (Figure 5h).

### 3.6. Renal Glycogen Analysis

To determine the possible role of isoflavone in glycogen modulation, we evaluated the PSA in real tissue (Figure 6). We evidenced that the DM1 and DM1 + OVX groups had increased renal tissue glycogen storage when compared with the control group (* *p* < 0.05 and **** *p* < 0.0001, respectively). In addition, increased glycogen storage was found in the untreated OVX group when compared with the untreated DM1 + OVX group (*** *p* < 0.001) (Figure 6g). However, treatment with isoflavone and 17β-estradiol in the DM1 + OVX group could not significantly reduce the amount of glycogen. 

## 4. Discussion

In this work, in a model of diabetes followed by ovariectomy, we demonstrated the role of isoflavone in protecting from renal injury via PPAR-γ. In our model, isoflavone-treated animals were protected from renal injury, observed by a reduction in the renal injury marker KIM-1 in the renal tissue. We postulate that this isoflavone-mediated protection of kidney injury occurs via PPAR-γ, where we observed that—mainly in the model of diabetes concomitant to ovariectomy—the endometrium as much as the kidney presents a reduction in PPAR-γ, observed at the proteinic and gene levels. Isoflavone treatment for this condition can positively modulate PPAR-γ at the protein and gene levels, demonstrating this possible relationship. 

The relationship of isoflavone with diabetes and menopause has been reviewed by numerous authors [25,26,27]. Induction of diabetes and ovariectomy exacerbates blood glucose [28]. In systematic reviews, there is a disparity of ideas on this. It has been reported that isoflavone can reduce the risk of type II diabetes mellitus [29,30], but, in another systematic review, isoflavone did not significantly interfere with glucose metabolism. In further work, it has been evidenced that extracted isoflavones and those from soy protein improve the lipid profile and may prevent cardiovascular events in diabetic individuals [31]. 

Isoflavone and 17β-estradiol negatively regulate blood glucose [7,24]. This similar effect of isoflavone or 17β-estradiol occurs due to estrogenic characteristics [7]. Investigations of our group have demonstrated the beneficial effect of isoflavone in ovariectomized diabetic animals [13,14]. Above all, in this current report, we demonstrate that isoflavone shows a similar effect to 17β-estradiol, leading to blood glucose homeostasis, but interestingly, the isoflavone was more effective in inducing protein synthesis of PPAR-_Y_ in renal tissue of diabetic animals than estrogen, and this promoted lower expression of KIM-1 in this condition. Thus, our data suggest that isoflavone was more effective in inhibiting the renal injury induced by hyperglycemia than the estrogen itself in our model. Therefore, these findings will be better explored in future studies.

In animals, PPAR-γ^−/−^ demonstrates cross-links of glucose and lipid metabolism between adipose tissue, muscle, and liver, suggesting that PPARγ is important for maintaining normal insulin sensitivity and whole-body glucose and lipid homeostasis [32]. Isoflavone is a potent agonist of PPARα/γ and exerts anti-inflammatory activity, which may contribute to the prevention of metabolic syndrome [33]. We have previously shown in a model of renal injury by a hyper-lipidic diet that isoflavone prevents renal injury via PPAR-γ [14]. 

In this work, we demonstrated that protein loss and gene expression of PPAR-γ in endometrium but not in the kidney of ovariectomized diabetic animals may be related to greater kidney injury due to a greater susceptibility to glucose. Furthermore, we demonstrated through transcriptome analysis in human proximal renal tubule cells that the reduction in PPAR-γ expression may be glucose-concentration-dependent. Higher concentrations of glucose (i.e., 16 mM) cause an expressive decline in PPAR-γ expression. We observed overexpression of KIM-1 in diabetic and ovariectomized diabetic animals, but isoflavone could reduce this marker of renal damage significantly only in the ovariectomized diabetic group. 

The fibrosis present in human diabetic nephropathy reduces renal function dramatically [34]. In diabetic animals induced by STZ, an accumulation of collagen and overexpression of pro-fibrotic genes are observed [35]. In our model, we observed that, especially in the ovariectomized diabetic group, this accumulation of collagen occurs when compared to the control group and ovariectomized-only group. Through transcriptome analysis, we demonstrated the overexpression of type I collagen, in particular, through the overexpression of the *COL1A1* gene. In our in vivo model, treatment with isoflavone or 17β-estradiol could maintain the collagen profile, which was unchanged when compared to the control. Calycosin, a type of isoflavone, could also reduce pro-fibrotic factors in a high-fat diet/STZ-induced T2DM model [36], an effect that was also observed in pulmonary fibrosis [37]. 

Diabetes induced by SZT and a high-fat diet increases PAS accumulation [38,39] This glycogen accumulation, observed by PAS, also occurs in humans with diabetic nephropathy [34,40]. Our data were in agreement with these observations, where we demonstrated that mainly diabetic-ovariectomized but also diabetic-only animals accumulate glycogen in the renal tissue in an exacerbated manner. Similarly to collagen accumulation, isoflavone and 17β-estradiol treatment maintained tissue glycogen similar to the control, even under diabetes induction. 

## 5. Conclusions

Through our observations, we can conclude that isoflavone protects against kidney damage induced by gonadal failure and DM1 through PPAR-γ expression. In addition, we can conclude that glucose is a potent negative regulator of PPAR-γ. Isoflavone stimulates PPAR-γ protein and gene expression. Thus, we postulate that the benefits observed by isoflavone treatment may occur via PPAR-γ. 

Further investigations of the downstream pathway of this receptor may allow new perspectives to be developed on therapeutic approaches for diabetic nephropathy, especially for postmenopausal women, which will benefit individuals affected by diabetes.

## Figures and Tables

**Figure 1 nutrients-14-02567-f001:**
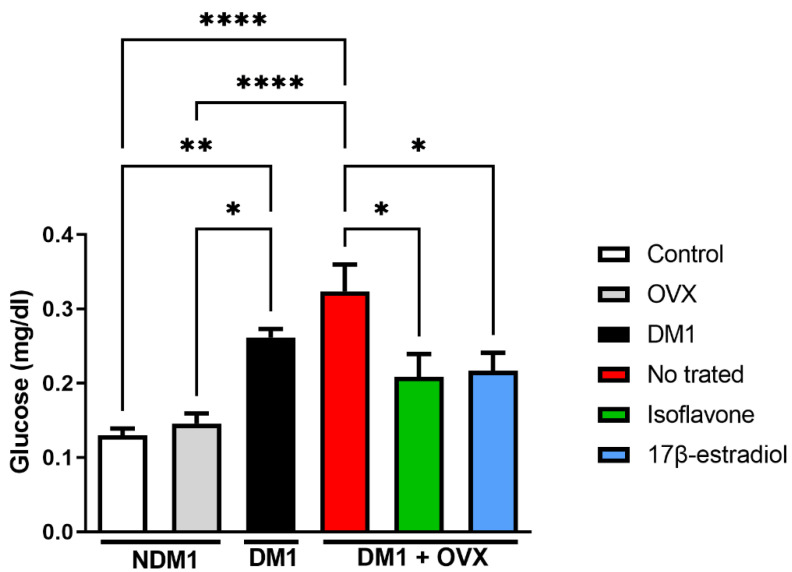
Isoflavone reduces glycemia. Blood glycemia was measured in mg/dL in non-diabetic, diabetic, and isoflavone- and 17β-estradiol-treated groups. Values represent mean ± SEM (*n* = 6). Kruskal–Wallis test was performed, followed by Dunn’s multiple comparison test (* *p* < 0.05; ** *p* < 0.01; **** *p* < 0.0001).

**Figure 2 nutrients-14-02567-f002:**
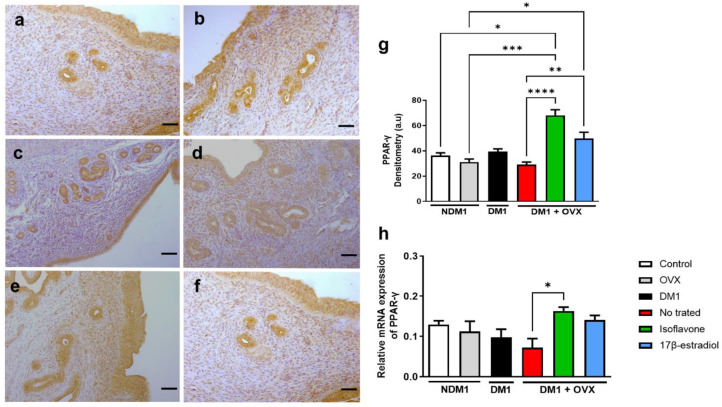
Isoflavone induces PPAR-γ in the endometrium. Immunoreactivity of PPAR-γ by immunohistochemistry (**a**–**f**). Control (**a**), OVX (**b**), DM1 (**c**), untreated DM1 + OVX (**d**), DM1 + OVX treated with isoflavone (**e**), and DM1 + OVX treated with 17β-estradiol (**f**). Densitometry analysis is represented in arbitrary units (**g**). Scale bar = 50 μm. Expression of PPAR-γ mRNA with β-actin gene as endogenous control. (**h**). Values represent mean ± SEM (*n* = 6). Kruskal–Wallis test was performed, followed by Dunn’s multiple comparison test (* *p* < 0.05; ** *p* < 0.01; *** *p* < 0.001; **** *p* < 0.0001).

**Figure 3 nutrients-14-02567-f003:**
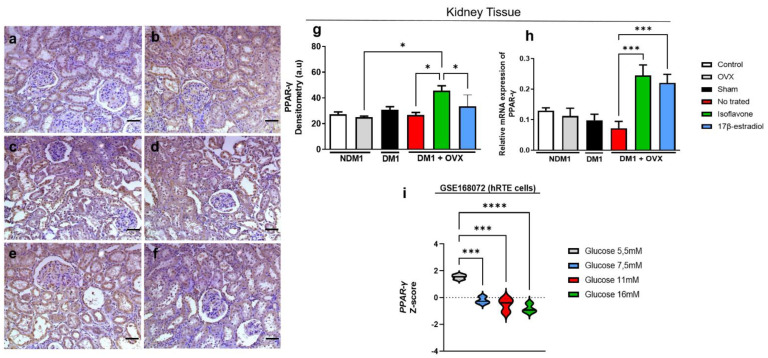
Isoflavone induces PPAR-γ in the kidneys. Immunoreactivity of PPAR-γ by immunohistochemistry (**a**–**f**). Control (**a**), OVX (**b**), DM1 (**c**), untreated DM1 + OVX (**d**), DM1 + OVX treated with isoflavone (**e**), and DM1 + OVX treated with 17β-estradiol (**f**). Densitometry analysis is represented in arbitrary units (**g**). Scale bar = 50 μm. mRNA expression of PPAR-γ, with B-actin gene as endogenous control. (**h**). Transcriptome analysis (**i**), Values represent mean ± SEM (*n* = 6). Kruskal–Wallis test was performed, followed by Dunn’s multiple comparison test for non-normal distribution of data, and one-way ANOVA followed by Dunnett’s multiple comparison test for a normal distribution of data (* *p* < 0.05; *** *p* < 0.001; **** *p* < 0.0001).

**Figure 4 nutrients-14-02567-f004:**
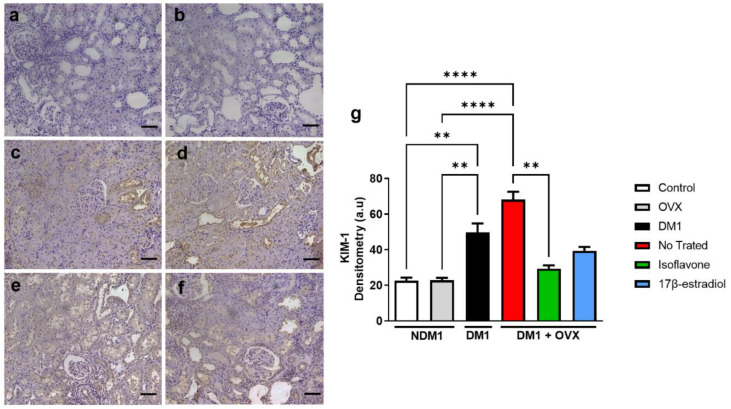
Isoflavone protects diabetic renal tissue of ovariectomized animals. Immunoreactivity of KIM-1 by immunohistochemistry (**a**–**f**). Control (**a**), OVX (**b**), DM1 (**c**), untreated DM1 + OVX (**d**), DM1 + OVX treated with isoflavone (**e**), and DM1 + OVX treated with 17β-estradiol (**f**). Densitometry analysis is represented in arbitrary units (**g**). Scale bar = 50 μm. Values represent mean ± SEM (*n* = 6). Kruskal–Wallis test was performed, followed by Dunn’s multiple comparison test for non-normal distribution of data. ** *p* < 0.01; **** *p* < 0.0001.

**Figure 5 nutrients-14-02567-f005:**
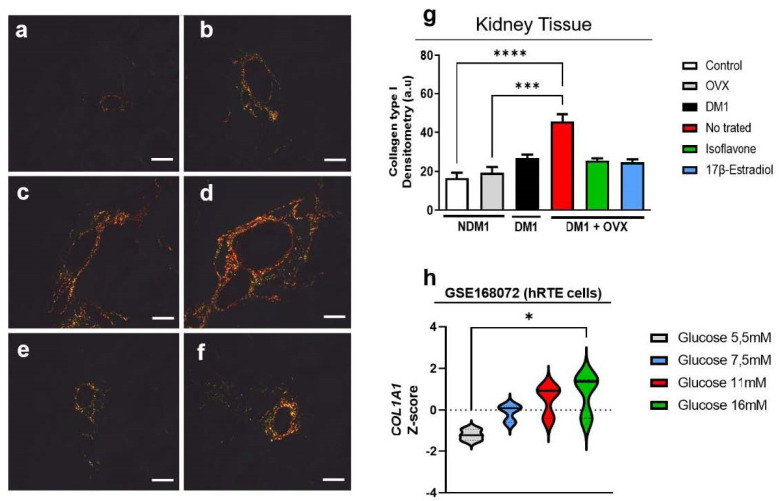
Isoflavone does not reduce collagen in renal tissue. Control (**a**), OVX (**b**), DM1 (**c**), untreated DM1 + OVX (**d**), DM1 + OVX treated with isoflavone (**e**), and DM1 + OVX treated with 17β-estradiol (**f**). Densitometry analysis is represented in arbitrary units (**g**). Scale bar = 50 μm. Transcriptome analysis (**h**). Values represent mean ± SEM (*n* = 6). Kruskal–Wallis test was performed, followed by Dunn’s multiple comparison test for non-normal distribution of data (* *p* < 0.05; *** *p* < 0.001 and **** *p* < 0.0001).

**Figure 6 nutrients-14-02567-f006:**
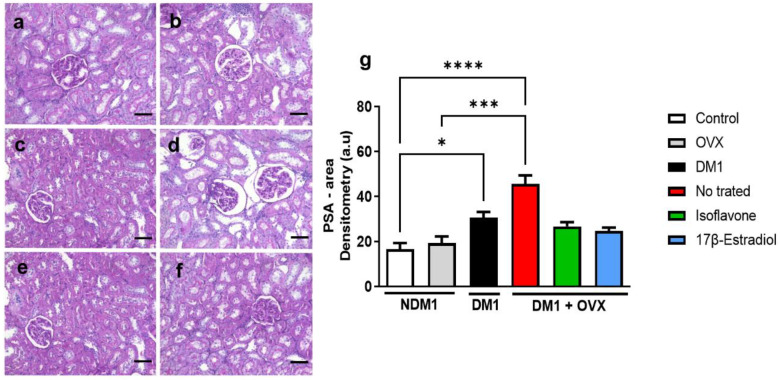
Isoflavone does not decrease glycogen storage in the actual tissue. Control (**a**), OVX (**b**), DM1 (**c**), DM1 + OVX untreated (**d**), DM1 + OVX treated with isoflavone (**e**), and DM1 + OVX treated with 17β-estradiol (**f**). Densitometry analysis is represented in arbitrary units (**g**). Scale bar = 50 μm. Values represent mean ± SEM (*n* = 6). Kruskal–Wallis test was performed, followed by Dunn’s multiple comparison test for non-normal distribution of data (* *p* < 0.05; *** *p* < 0.001; **** *p* < 0.0001).

**Table 1 nutrients-14-02567-t001:** The sequence of primer pairs for RT-qPCR.

Gene	Forward	Reverse
*PPAR-γ*	GGAGCCTAAGTTTGAGTTTGCTGTG	TGCAGCAGGTTGTCTTGGATG
*β-actin*	GGAGATTACTGCCCTGGCTCCTA	GACTCATCGTACTCCTGCTTGCTG

## Data Availability

Data are available upon request.

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
