# Peer review of "Isoflavone Protects the Renal Tissue of Diabetic Ovariectomized Rats via PPARγ"

_nutrients, 2022, doi:10.3390/nu14132567_

Round 1

Reviewer 1 Report

Work by Carbonel et al. on "Isoflavone protects the renal tissue of diabetic ovariectomized rats via PPAR-γ" is an extremely interesting and well-written literature. It only requires a few necessary fixes:

- in the part introduction there is no clearly defined research purpose of this publication. We have information about the research carried out by the team recently, but there are really no two specific sentences about the very purpose of the research presented below. The sentence "(...) Taking into account the protective function of PPAR-γ in kidney injury [15–17], we evaluated the contribution of soy isoflavone via PPAR-γ in an experimental animal model of menopause and type 1 diabetes mellitus." it does not introduce the reader to the purpose of this work.

- in the materials and methods section, more precisely in subsection 2.6, please add the source links provided to the literature and leave only the numbers in the text (references to the bibliography).

- please put the caption for figure 1 directly below the figure, not above it. In other cases, throughout the text, the descriptions of the figures appear below the figure itself.

- I am asking you to put the scale bare in all photos (a-f) in figure 2. Additionally, I am asking you to increase the graph which is part of the g figure, because in its current form the scale is very poorly visible and it makes it difficult to receive the presented results.

- please put the scale bare in all photos (a-f) in figure 3. In addition, I am asking you to increase the graph representing part h and i of the figure, because in its current form the scale is very poorly visible and it makes it difficult to receive the presented results.

- I also have a question, is it possible to standardize the scale between the presented experiments? I mean the graphs of mRNA expression of PPAR-γ (Figure 2h and Figure 3 h) and for Densitometry analysis is represented in arbitrary units (Figure 3g and Figure 2 g).

- please put the scale bare in all photos (a-f) in figures 4, 5 and 6.

- please remove the numbering (6) from the Reference header. According to the form, this part has no numbering.

Author Response

REVISOR 1

Work by Carbonel et al. on "Isoflavone protects the renal tissue of diabetic ovariectomized rats via PPAR-γ" is an extremely interesting and well-written literature. It only requires a few necessary fixes:

- in the part introduction there is no clearly defined research purpose of this publication. We have information about the research carried out by the team recently, but there are really no two specific sentences about the very purpose of the research presented below. The sentence "(...) Taking into account the protective function of PPAR-γ in kidney injury [15–17], we evaluated the contribution of soy isoflavone via PPAR-γ in an experimental animal model of menopause and type 1 diabetes mellitus." it does not introduce the reader to the purpose of this work.

 Answer:

Dear Revisor,

We appreciate the feedback; it’s are important for the development of our investigation. We rewrote the part and introduced the reader to the purpose of our study by modifying paragraph 3, starting from the 10th line (in red) as follows:

“.... the objective of this work was to evaluate the effect of ovarectomy and the consequent gonadal failure on the development of DM1-induced kidney injury, as well as the contribution of soy isoflavone via PPAR-γ in this experimental animal model. Our hypothesis is that gonadal failure potentiates renal injury induced by chronic hyperglycemia and that isoflavone treatment attenuates this condition by PPAR-γ induction.  

Our data revealed that isoflavone protects the kidney against gonadal failure associated with DM1 damage. Increased protein and gene expression of PPAR-γ was evidenced under the influence of isoflavone. In addition, we demonstrate for the first time that PPAR-γ reduction occurs dose-dependent on glucose. On the other hand, we have not clearly demonstrated the consequences of ovarectomy in the renal injury development and other findings are necessary to confirm this hypothesis.”

 in the materials and methods section, more precisely in subsection 2.6, please add the source links provided to the literature and leave only the numbers in the text (references to the bibliography).

 Answer: The reference [24] was added. S. Shrestha et al., “Elevated glucose represses lysosomal and mTOR-related genes in renal epithelial cells composed of progenitor CD133+ cells,” PLoS One, vol. 16, no. 3 March, pp. 1–21, 2021, doi: 10.1371/journal.pone.0248241.

- please put the caption for figure 1 directly below the figure, not above it. In other cases, throughout the text, the descriptions of the figures appear below the figure itself.

Answer: The caption of Fi[gure 1 was placed below the image

- I am asking you to put the scale bare in all photos (a-f) in figure 2. Additionally, I am asking you to increase the graph which is part of the g figure, because in its current form the scale is very poorly visible and it makes it difficult to receive the presented results.

Answer: the scale bare in all photos (a-f) was added e and the size of the graph in figure g was increased

- please put the scale bare in all photos (a-f) in figure 3. In addition, I am asking you to increase the graph representing part h and i of the figure, because in its current form the scale is very poorly visible and it makes it difficult to receive the presented results.

 Answer: the scale bare in all photos (a-f) was added e and the size of the graph in figure h and i was increased

I also have a question, is it possible to standardize the scale between the presented experiments? I mean the graphs of mRNA expression of PPAR-γ (Figure 2h and Figure 3 h) and for Densitometry analysis is represented in arbitrary units (Figure 3g and Figure 2 g).

Answer: All the scale in the statistical analysis charts in this paper have been standardised.

 - please put the scale bare in all photos (a-f) in figures 4, 5 and 6.

Answer: the scale bare in all photos (a-f) was added

please remove the numbering (6) from the Reference header. According to the form, this part has no numbering.

 Answer: This has been removed

Reviewer 2 Report

Carbonel et al. in the manuscript investigate the protective properties of isoflavones to kidneys in diabetic and post-menopause conditions.  The results presented in the paper are interesting and promising. However, the introduction must be improved. The introduction should define the purpose of the work and its significance, including specific hypotheses being tested. Also, the introduction should briefly mention the main aim of the work and highlight the main conclusions. All of this is currently missing. 

The authors start with one sentence about DM1 and in a later paragraph write about DM2. What is the risk of DM1 vs DM2 in post-menopause women? Are there any reports of isoflavones and DM1? Somewhere in the text, it should be clearly stated which diabetes the experimental model mimics. 

Interesting that in some cases the effect of isoflavones was greater than that of estradiol. Do the authors have an explanation/hypothesis for this?

My specific comments are listed below:

-       Section 2.1: Was a sham surgery performed? What tissues and how were they obtained? How were they embedded in paraffin? How were the tissues stored/preserved for RNA isolation?

-       Section 2.1: isoflavones and estrogen treatments were daily or one time? Where did the reagents come from?

-       Section 2.3: “The intensity of collagen fibers’ birefringence was measured as previously reported [20].” - please describe briefly

-       Section 2.5: What amount of RNA was taken for cDNA synthesis? How was the amount of RNA determined? How was RNA checked for protein/DNA contamination? What amount of tissue was taken for isolation?

-       Section 2.6: “Human renal proximal tubule epithelial cells” - was it a primary or immortalized line? How was the culture performed? How long were the cells exposed to glucose? How often was the medium changed to ensure the effect of constant exposure to a given concentration?

-       Section 2.6: all default options of GEO2R were used?

-       Figure 2: no tissue localization in the figure description

-       Figure 3: in the graph description it should be clearly indicated which results are derived from tissues and which from cells

-       Figure 4: 

-       Figure 6: reading would be much easier if the captions of the figure were on the same page as the figure

-       Section 3.6: no glycogen analysis in the M&M section, PSA - no explanation of the abbreviation

-       Section 4: “in a model of diabetes followed by ovariectomy” - In the materials and methods it is the other way round - the first ovariectomy then diabetes, so what was the order?

-       Section 4: “...injury marker KIM-1 and inhibition of glycogen and type I collagen accumulation in the....” – according to fig.6 isoflavone does not decrease glycogen storage

-       Section 4: “In this work, we demonstrated that the loss of protein and gene expression of PPAR-γ in the kidney and endometrium of ovariectomized diabetic animals...” - There were no statistically significant differences in PPAR levels between DM1+OVX and control

Author Response

Carbonel et al. in the manuscript investigate the protective properties of isoflavones to kidneys in diabetic and post-menopause conditions.  The results presented in the paper are interesting and promising. However, the introduction must be improved. The introduction should define the purpose of the work and its significance, including specific hypotheses being tested. Also, the introduction should briefly mention the main aim of the work and highlight the main conclusions. All of this is currently missing. 

The authors start with one sentence about DM1 and in a later paragraph write about DM2. What is the risk of DM1 vs DM2 in post-menopause women? Are there any reports of isoflavones and DM1? Somewhere in the text, it should be clearly stated which diabetes the experimental model mimics. 

Interesting that in some cases the effect of isoflavones was greater than that of estradiol. Do the authors have an explanation/hypothesis for this?

Answer:

Dear Revisor,

We appreciate the feedback; it’s are important for our development as the investigation. We had the text part rewritten and introduced the reader to the purpose of the study by modifying paragraph 3, starting from the 10th line (in red) as follows:

….The objective of this study was to evaluate the effect of ovariectomy and the consequent gonadal failure on the development of DM1-induced kidney injury, as well as the contribution of soy isoflavone via PPAR-γ in this experimental animal model. Our hypothesis is that gonadal failure potentiates renal injury induced by chronic hyperglycemia and that isoflavone treatment attenuates this condition by PPAR-γ induction.  

Our data revealed that isoflavone protects the kidney against gonadal failure associated with DM1 damage. Increased protein and gene expression of PPAR-γ was evidenced under the influence of isoflavone. In addition, we demonstrate for the first time that PPAR-γ reduction occurs dose-dependent on glucose. On the other hand, we have not clearly demonstrated the consequences of ovariectomy in the renal injury development and other findings are necessary to confirm this hypothesis.

We inserted at the end of the paragraph 2º the choice of type 1 diabetes model “Nevertheless, few studies have demonstrated the effect of isoflavone on T1D, especially in renal injury [Guo TL et al., 2014Stephenson et al., 2005; Von  Hertzen et al., 2004]” We complement the paragraph 1º of the conclusion according to the hypothesis request “Through our observations, we can conclude that isoflavone protects kidney damage induced by gonadal failure and DM1 through PPAR-γ expression. In addition, we can conclude that glucose is a potent negative regulator of PPAR-γ”

Interesting that in some cases the effect of isoflavones was greater than that of estradiol. Do the authors have an explanation/hypothesis for this?

Answer: Our hypothesis was described complementing paragraph 3º the isoflavone was more effective in inducing protein synthesis of PPAR-Y in renal tissue of diabetic animals than estrogen and this promoted lower expression of KIM-1 in this condition. Thus, our data suggest that isoflavone was more effective in inhibiting the renal injury induced by hyperglycemia than the estrogen itself in our model. Therefore, these findings will be better explored in future studies

My specific comments are listed below:

Section 2.1: Was a sham surgery performed? What tissues and how were they obtained? How were they embedded in paraffin? How were the tissues stored/preserved for RNA isolation?

Answer: We inserted in Section 2.1: A portion of the endometrium and kidney was immediately immersed in liquid nitrogen e and then stored in the -80ºC freezer for qPCR assay. Another portion of the endometrium and kidney were immersed in paraformaldehyde 4% for histological processing.

Section 2.1: isoflavones and estrogen treatments were daily or one time? Where did the reagents come from?

Answer: We inserted in Section 2.1: Treatment with soy isoflavone and 17β-estradiol was performed for 30 days after T1D induction.

Soy isoflavones NovasoyVR (Archer Daniels Midland, Decatur, IL) containing 40% of total isoflavones in the ratio of 1.3:1:0.3 genistein: daidzein: glycitein, respectively, 7–12% protein, 4% ash and 6% moisture, and the remaining 41% consists of the other soy phytocomponents. In addition, we used 17β-estradiol (Sigma-Aldrich).

  Section 2.3: “The intensity of collagen fibers’ birefringence was measured as previously reported [20].” - please describe briefly

Answer: We inserted in Section 2.1: “PS-stained kidneys sections were also analyzed under polarized light to differentiate collagen fibers type I (yellow and red).  For that, randomly selecting four nonoverlapping areas in each slide were considered at 40X magnification using Axiolab 2.0 optical micro-scope (Zeiss, Germany). The quantification of the area occupied by the collagen fibers was performed using the Image J software (NIH, Baltimore). Densitometry analysis data were obtained as arbitrary units between 0 and 255. A mean of values from the two sec-tions/animals was obtained to calculate the mean of each group. (Koshimizu et al. 2013).”

Section 2.5: What amount of RNA was taken for cDNA synthesis? How was the amount of RNA determined? How was RNA checked for protein/DNA contamination? What amount of tissue was taken for isolation?

Answer: We have rewritten the methodology to facilitate understanding:

“According to the manufacturer’s instructions, total RNA from the endometrium and kidney was isolated using TRIzol Reagent (Thermo Fisher Scientific). Total RNA was treated with DNase (RQ1 RNase-Free DNase, Promega) to prevent genomic deoxyribonucleic acid contamination and the RNA pellet was resuspended in RNase-free water. For mRNA expression analysis, total RNA (1 μg) of the endometrium and kidney samples was reverse transcribed using SuperScript III RT (Thermo Fisher Scientific) according to the manufacturer's protocol. The PCR product was amplified from cDNA using QuantiFast SYBR Green PCR kit (Qiagen) and specific primers for rat (Table 1)

 Section 2.6: “Human renal proximal tubule epithelial cells” - was it a primary or immortalized line? How was the culture performed? How long were the cells exposed to glucose? How often was the medium changed to ensure the effect of constant exposure to a given concentration?

Answer: Primary human renal tubular epithelial (hRTE) cells. The cells were grown using serum-free condition. The growth formulation consisted of a 1:1 mixture of Dulbecco’s modified Eagles’ medium and Ham’s F-12 growth medium supplemented with selenium (5 ng/ml), insulin (5 μg/ml), transferrin (5 μg/ml), hydrocortisone (36 ng/ml), triiodothyronine (4 pg/ml), and epidermal growth factor (10 ng/ml). The cells were fed fresh growth medium every 3 days, and at confluence, the cells were sub-cultured using trypsin–ethylenediaminetetraacetic acid (0.05%, 0.02%). For use in experimental protocols, cells were subcultured at a 1:2 ratio, allowed to reach confluence (7 days following subculture) and fed with 5.5, 7.5, 11 or 16 mM glucose for 7 days followed by sub-culturing the cells for 3 passages (P3) in media containing 5.5, 7.5, 11 or 16 mM glucose. Typically, the cells were passaged, 1:2 for three passages before the initiation of the experimente.

Section 2.6: all default options of GEO2R were used?

Answer: GEO2R was used to screen for genes differentially expressed in each group.

Figure 2: no tissue localization in the figure description

Answer: It was added in the legend of figure 2 - Isoflavone induces PPAR-γ in indometrium

Figure 3: in the graph description it should be clearly indicated which results are derived from tissues and which from cells

Answer: Figure 2 and 3 shows the data generated from animal tissue and data generated from cells.

Figure 6: reading would be much easier if the captions of the figure were on the same page as the figure

Answer: We leave the pictures with the legend to make it easier to read.

Section 4: “in a model of diabetes followed by ovariectomy” - In the materials and methods it is the other way round - the first ovariectomy then diabetes, so what was the order?

Answer: We reported in the section 2.1:

Induction of type 1 diabetes was performed one day after ovariectomy.

Section 4: “...injury marker KIM-1 and inhibition of glycogen and type I collagen accumulation in the....” – according to fig.6 isoflavone does not decrease glycogen storage

Answer: The treatment with isoflavone and 17β-estradiol in the DM1+OVX group could not significantly reduce the amount of glycogen.

The affirmation “and inhibition of glycogen and type I collagen accumulation”, it was withdrawn from the discussion.

Section 4: “In this work, we demonstrated that the loss of protein and gene expression of PPAR-γ in the kidney and endometrium of ovariectomized diabetic animals...” - There were no statistically significant differences in PPAR levels between DM1+OVX and control

Answer: In this work, we demonstrated that protein loss and gene expression of PPAR-γ in endometrium, but not in kidney of ovariectomized diabetic.  Animals may be related to greater kidney injury due to a greater susceptibility to glucose.

The affirmation “A reduction in it may be associated with renal tissue damage as well as a greater accumulation of glycogen and collagen” it was withdrawn from the conclusion.
